# Anti-Aging Effects of R-Phycocyanin from *Porphyra haitanensis* on HUVEC Cells and *Drosophila melanogaster*

**DOI:** 10.3390/md20080468

**Published:** 2022-07-22

**Authors:** Yanyu Feng, Hanjin Lu, Jiamiao Hu, Baodong Zheng, Yi Zhang

**Affiliations:** 1Engineering Research Centre of Fujian-Taiwan Special Marine Food Processing and Nutrition, Ministry of Education, Fuzhou 350000, China; fengyanyu1221@163.com (Y.F.); lu147569@163.com (H.L.); jiamiao.hu@fafu.edu.cn (J.H.); 2College of Food Science, Fujian Agriculture and Forestry University, Fuzhou 350000, China

**Keywords:** *Porphyra haitanensis*, R-phycocyanin, HUVEC cells, *Drosophila melanogaster*, anti-aging

## Abstract

Aging has become a global public health challenge. Many studies have revealed that the excessive generation of ROS and oxidative stress could be the major causative factors contributing to aging. In this study, R-phycocyanin (R-PC) was isolated from *Porphyra haitanensis*, and its anti-aging ability was explored by natural aging *Drosophila melanogaster* and H_2_O_2_-induced HUVEC cells as the aging model. Results showed that R-PC α and β subunits expressed have antioxidant activity and can inhibit the generation of radicals, exhibiting a protective effect against H_2_O_2_-induced apoptotic HUVEC cells death. R-PC prevented the H_2_O_2_-induced HUVEC cell cycle phase arrest by regulating cell cycle-related protein. Furthermore, R-PC prevented the H_2_O_2_-induced HUVEC cell cycle phase arrest by regulating cell-cycle-related protein expression. In vivo study also indicated that R-PC significantly increased the survival time and alleviated the oxidative stress of *Drosophila melanogaster*. Moreover, R-PC notably decreased levels of ROS in natural aging flies and inhibited lipid peroxidation by enhancing the expressions of the endogenous stress marker genes (SOD1, SOD2, CAT of *Drosophila melanogaster*). Taken together, a study on the antioxidation extract from *Porphyra haitanensis*, such as R-PC, may open a new window for the prevention of anti-aging.

## 1. Introduction

The world is in the midst of an aging population challenge, which means moving from a world in which the majority of the population is relatively young to one in which a significant proportion of people are over the age of 65. At present, numerous older people will devote themselves to keeping a healthy body. However, this segment of the population is subject to physical and cognitive impairment at higher rates than younger people [1,2]. Older people can improve their quality of life by supplying anti-aging ingredients in their diet.

A gradual decline in functional organ reserves is characterized by aging. It decreased the ability to maintain homeostasis under conditions of stress. The accrual of oxidative stress and DNA damage has been the predominant mechanistic explanation for the process of aging for many years [3]. Aging induces changes in the morphology and functions of the tissues and organs of the body change with age, which is closely related to oxidative stress [4]. Oxidative stress occurs due to the relatively imbalanced state between production and elimination of ROS in the body. It causes the decline of the body’s antioxidant stress system and various aging-related diseases, such as atherosclerosis, hypertension, arthritis, and neurodegenerative disorders [5]. Mitochondria are major determinants of cell life and apoptosis. In normal physiological conditions, people’s bodies have internal antioxidant enzyme defense systems, such as SOD, CAT, and GSH-Px, and the production and cleaning of free radicals are in a dynamic balance [6,7]. However, this balance is broken when the external environment stimulates the dynamic balance, and the free radicals accumulate in large quantities, leading to mitochondria injury, which further causes cell death and metabolism disorders, causing damage to cells and tissues [8]. Antioxidants can slow the damage caused by oxidative stress and regulate numerous signaling pathways that control a wide range of cellular processes, including cell cycle and apoptosis. Thus, supplementation with natural antioxidants in diets is a relatively easy and valid way to dominate oxidative stress.

In China, *Porphyra haitanensis* is a sort endemic and the first to be artificially cultivated. It has had the most considerable biomass output among many *Porphyra* crops in China, Japan, and South Korea [9]. Recently, many natural compounds from Porphyra haitanensis have shown excellent activity against aging in vitro and in vivo. Phycocyanin is a light-harvesting protein–pigment complex affiliated to the phycobilisome cores and a significant constituent of *Porphyra haitanensis*. Phycocyanins are divided into sub-categories according to their sources and spectral characteristics, supplemented by the prefixes R-, B-, C-, etc. Currently, C-phycocyanin has significant radical-scavenging activity, in addition to its utilization as a natural colorant in the food industry. R-phycocyanin (R-PC) is a biliprotein pigment with an open-chain tetra pyrrole chromophore and has various critical biological properties [10,11]. The study reported that R-PC from *Porphyra haitanensis* has excellent antioxidant and radical-scavenging characteristics, thereby protecting against antioxidative stress.

In recent years, oxidative stress has been one of the most critical mechanisms of cellular senescence, and H_2_O_2_ has been the most commonly used inducer for stress-induced premature senescence (SIPS) in cells, which shares features with replicative senescence [12,13]. At present, H_2_O_2_-induced cell models have identified several natural anti-aging active substances, such as C-phycocyanin and resveratrol [14,15]. Meanwhile, the molecular pathways that determine the aging process and prolong life have been deeply studied using animal models, and significant progress has been made. However, it is essential that data obtained from a mouse model may not extrapolate directly to natural aging, and more comprehensive research is needed in a natural aging model [16]. *Drosophila melanogaster* has a short lifespan, fast reproduction, easy feeding conditions, and similar metabolic pathways and aging genes to people’s bodies [17,18]. This study aimed to investigate the link between R-PC structures and antioxidation, and the potential anti-aging effects of R-PC were investigated on cell apoptosis, cycle, *Drosophila melanogaster* lifespan, level of ROS, and critical molecular proteins.

## 2. Results

### 2.1. Isolation and Purification of R-PC

As shown in Figure 1A, the phycobili protein extract was separated by Sephadex G-150 gel chromatography with the R-PC absorbance at 280 nm, 499 nm, 545 nm, and 615 nm. The characteristic absorption peaks of phycoerythrin decreased significantly, while the characteristic absorptions of R-PC increased significantly at 545 nm and 615 nm. The result indicated that a large amount of phycoerythrin and some leucoproteins, which were absorbed in 280 nm, had been removed by Sephadex G-150 gel chromatography. However, a weak phycoerythrin characteristic absorption can still be observed at 499 nm, and the ratio of A_615_/A_499_ was about 3, indicating that the obtained R-PC sample still contained a small amount of RPE, which needed further purification. After purification by DEAE-Sepharose FF chromatography, the obtained R-PC sample only showed the characteristic absorption peaks of R-PC at 545 nm and 615 nm, and the ratio of A_615_/A_499_ increased to 8, indicating that R-PC was further purified.

XQ Chen et al. showed that SDS-PAGE detected the relative molecular weights of subunits consisting in R-PC, α/β subunit is about 18–20 kDa, and γ about 35 kDa. Previous reports showed that R-PC was isolated and purified from the phycobiliprotein extract was composed of an α subunit and four β subunits (β_20.0_^5.5^, β_20.9_^5.5^, β_20.0_^5.6^, β_20.9_^5.6^) with different PIs and molecular mass, so the resulting R-PC there should be β subunits composed of different trimer forms [19,20,21]. Similarly, in this study, the SDS-PAGE and denaturing IEF results demonstrated that the R-PC was proved to have three subunits (18.5 kD α (α_18.5_), 21.2 kD β (β_21.2_), 23.9 kD β (β_23.9_)), and PI of them are 6.4, 5.6 and 5.5, respectively (Figure 1B). PH range from 5.0 to 8.0 in denaturing-IEF, R-PC exhibited a stable subunit band at pH 6.5 (α^6.5^) (Figure 1C); PH range from 4.0 to 6.5 in denaturing-IEF (Figure 1D), R-PC exhibited two stable subunit bands at PH 5.5 (β^5.5^) and 5.6 (β^5.6^). These findings suggested that R-PC from the porphyra haitanensis extract is composed of one α subunit (α_18.5_^6.5^) and four β subunits (β_21.2_^5.5^, β_23.9_^5.5^, β_21.2_^5.6^, β_23.9_^5.6^) with different PIs and molecular weights. Therefore, R-PC has various trimer forms of β subunits, and these β subunits are composed of different trimers in molecular size (or molecular mass), charge the difference in structural properties such as mass ratio and pI is very small, which is similar to the results of XQ et al. Many studies showed that these structures had the close link with antioxidation. Then we test whether antioxidant activity in R-PC.

### 2.2. Antioxidant Capacity In Vitro

The ABTS, DPPH, and FRAP assays gave comparable results for the antioxidant effect measured in R-PC (Figure 2). In the FRAP method, the reducing power was also detected in phycocyanins and showed an important antioxidant activity in the FRAP assay, especially R-PC with a value equal to (IC_50_ = 2.61 ± 0.03 mg/mL), which is close to the value obtained by C-phycocyanin used as control (IC_50_ = 2.83 ± 0.11 mg/mL). A similar trend was also observed for the ABTS test: R-PC yielded the antioxidant ability with a value of IC_50_ equal to 0.13 ± 0.15 mg/mL, compared to the C-phycocyanin (IC_50_ = 0.28 ± 0.45 mg/mL). Furthermore, phycocyanins showed an important antioxidant activity, the scavenging ability of different phycocyanins against superoxide anion free radicals, especially R-PC extract with a value equal to (IC_50_ = 1.07 ± 0.31 mg/mL), which is close to the value obtained by C-phycocyanin used as control (IC_50_ = 2.12 ± 0.03 mg/mL). Antioxidant activity can reduce the damage caused by oxidative stress, and aging is closely associated with oxidative stress. Our results suggested that R-PC from *P**orphyra haitanensis* revealed interested antioxidant effects in these assays.

### 2.3. R-PC Inhibits H_2_O_2_-Induced Cytotoxicity in HUVEC Cells

To assess whether the cytotoxic activity of R-PC against HUVEC cells, cells were respectively incubated with 0, 1, 3, 10, 30, and 100 mg/mL resveratrol or R-PC, after which cell viability was investigated by the CCK-8 assay. R-PC is not cytotoxic since doses up to 100 mg/mL did not alter the cell viability in HUVEC cells (Figure 3A). The study showed that L02 cells were damaged by H_2_O_2_, with cell viability of only 40%. However, the C-phycocyanin groups restored cell viability. The more the C-phycocyanin concentration was, the better the restoration became. MDCK cells treatment with 20 mmol/L C-phycocyanin remarkably enhanced the viability of MDCK cells following 24 and 48 h treatment to oxalate [22,23]. In this study, to investigate whether R-PC has protection against oxidative stress, HUVEC cells were pretreated with resveratrol or R-PC for 24 h, then incubated by H_2_O_2_ for 30 min, and cell viability was explored through CCK-8 assay. The viability of HUVEC cells was reduced markedly through 100 µM H_2_O_2_ for 30 min, while R-PC protected cells from H_2_O_2_-induced cytotoxicity in a concentration-dependent manner (10 mg/mL, 54.6 ± 4.1%; 20 mg/mL, 59.8 ± 5.0%; 40 mg/mL, 71.2 ± 7.6%), as shown in Figure 3B. R-PC could markedly enhance the cell viability at 40 mg/mL (*p* < 0.001 versus the H_2_O_2_ alone group). In addition, SA-β-gal is a commonly used biomarker for aging cells, so we detected whether R-PC protects cells aged after H_2_O_2_ treatment by SA-β-gal staining. In addition, SA-β-gal is commonly regarded as an aging cell positive indicator, so this study detected whether R-PC protects cells aged after H_2_O_2_ treatment by SA-β-gal staining (Figure 3B). After treatment with H_2_O_2_, the number of positive cells with blue particles increased, while R-PC protected cells from H_2_O_2_-induced cells aged in a concentration-dependent manner (Figure 3C). These results showed that R-PC could protect HUVEC cells from oxidative stress relative to cellular injuries.

### 2.4. P-RC Protects against H_2_O_2_-Induced Apoptotic Cell Death in HUVEC Cells

In order to investigate whether R-PC decreases the apoptosis of HUVEC cells caused by H_2_O_2_, cells were treated with R-PC, Hoechst/PI staining, flow cytometry, JC-1 assay, and western blot were used to detect apoptosis. A microscopic fluorescence assay revealed the uniform amorphous of normal HUVEC cell’s nucleus and well-distributed deep blue fluorescence. H_2_O_2_-induced apoptosis cells showed typical apoptotic morphology, including broken membranes exhibiting nuclear red fluorescence in cells (Figure 4A). Resveratrol, a natural compound extracted from the skins of grapes, berries, or other fruits, has been shown to have anti-aging, anti-apoptotic, and antioxidative effects. R-PC also revealed similar anti-apoptotic effects, indicating that R-PC has a direct anti-apoptotic activity. R-PC could improve the apoptosis amorphous of HUVEC cells induced by H_2_O_2_. To quantitatively gain insight into the anti-apoptotic effects of R-PC in H_2_O_2_-induced HUVEC cells, after treatment with H_2_O_2_, the apoptosis rate of HUVEC cells was measured by Annexin-V/PI staining. As shown in Figure 4B, the apoptosis rate grew from 6.09 ± 0.4% to 75.69 ± 2.2% (versus the untreated group). By contrast, resveratrol (10 mg/mL, 20 mg/mL, 40 mg/mL) could evidently attenuate the apoptosis of HUVEC cells to 23.44 ± 1.23%, 35.89 ± 0.31%, 42.34 ± 3.24%, respectively. R-PC (10 mg/mL, 20 mg/mL, 40 mg/mL) could remarkably attenuate the apoptosis of HUVEC cells to 54.05 ± 4.6%, 49.09 ± 2.7% and 46.68 ± 3.1%, respectively. In which 40 mg/mL R-PC reduced the apoptosis remarkably (versus H_2_O_2_ alone group). Several studies showed that C-phycocyanin has beneficial effects on the development of porcine parthenos by attenuating mitochondrial dysfunction and oxidative stress. C-phycocyanin prevented the H_2_O_2_-induced compromise of mitochondrial membrane potential, release of cytochrome c from the mitochondria, and ROS generation [24,25]. Thus, it is to explore whether the inhibition of mitochondrial disruption may relate to the anti-apoptotic effect of R-PC and its impact on mitochondrial permeability in the presence of H_2_O_2_ using the JC-1 assay. The normal HUVEC cells tested by JC-1 revealed red fluorescence. The ΔΨm of mitochondrial was depolarized after treatment with H_2_O_2_, as shown through the decrease in the red fluorescence and the increase in green fluorescence. R-PC could restore the ΔΨm as indicated by increasing red fluorescence dose-dependently and repressing the green fluorescence. A total of 40 mg/mL R-PC treatment significantly ameliorated unbalanced the ΔΨm of mitochondrial induced by H_2_O_2_ in Figure 4C. In addition, western blot analyzed the expression of Bax, Bcl-2, caspase-3, and PARP in the HUVEC cells. As a result, R-PC significantly decreased Bax/Bcl-2 expression levels, and R-PC suppressed the activation of caspase-3 and PARP compared to cells exposed to H_2_O_2_ only (Figure 4D). These results indicate that R-PC exhibited a protective effect against H_2_O_2_-induced apoptotic HUVEC cells’ death.

### 2.5. R-PC Prevents H_2_O_2_-Induced Cell Cycle Progression at the G2/M Phase in HUVEC Cells

The entry of quiescent cells into the cell cycle is a critical event in cell growth. In order to evaluate whether R-PC influenced cell cycle progression in HUVEC cells, cells were treated with R-PC, and then the cell cycle progression of H_2_O_2_-treated HUVEC cells in the absence of R-PC using flow cytometry and western blot. The percentage of HUVEC cells in the S and G2/M phases were increased by treatment with H_2_O_2_ compared to treatment with control (S phase: 6.33 ± 3.2% in control vs. 12.72 ± 3.4% in H_2_O_2_, and G2/M: 23.16 ± 3.8% in control vs. 27.91 ± 2.2% in H_2_O_2_). The percentages of cells in the G0/G1 phase were reduced (68.37 ± 3.7% in control vs. 49.97 ± 2.9% in H_2_O_2_), indicating that H_2_O_2_ induced cell cycle arrest at the G2/M phase. In contrast, the percentages of both S-and G2/M-phase HUVEC cells treated with R-PC were slightly decreased compared with the H_2_O_2_-treated cells (S phase: 31.5 ± 3.4% in H_2_O_2_ vs. 25.9 ± 3.4% in R-PC, and G2/M: 10.3 ± 3.2% in H_2_O_2_ vs. 6.1 ± 4.0% in R-PC). The effects of H_2_O_2_ on HUVEC cells cycle progression were blocked by treatment with 40 mg/mL R-PC (G0/G1: 68.16 ± 3.8% in R-PC vs. 49.97 ± 4.2% in H_2_O_2_; S phase: 7.4 ± 3.4% in R-PC vs. 12.72 ± 3.9% in H_2_O_2_, and G2/M: 21.64 ± 4.0% in R-PC vs. 27.9 ± 3.6% in H_2_O_2_) (Figure 5A). Cell cycle progression is regulated by an ordered sequence of events that includes the activation of the cyclin-CDK. Activation of the CDKs requires their association with cyclins, whose levels fluctuate during the cell cycle. CDKs can interact with a group of proteins collectively termed cyclin-CDK inhibitors, such as p21 and p27 [26]. Research has shown that TGF-β1 induced cell cycle arrest in the S phase and G2/M phase, and C-phycocyanin lowered the percentage of cells in the S phase (9.30%) and G2/M phase (12.08%) in favor of G0/G1 phase (77.33%). The traverse of S and entry into the G2/M phase are controlled by the sequential activation of complexes containing the cyclins B1 and cdk1/2 [27]. As shown in Figure 5B, CDK1/2 and Cyclin B1 protein levels were suppressed in H_2_O_2_-treated HUVEC cells compared to cells treated with control, but p21 and p27 expression were enhanced. The enhanced p21 and p27 expressions were suppressed by treatment with R-PC. The suppressed cyclin D1 and CDK1/2 expression were also enhanced. These data suggested that R-PC prevents the H_2_O_2_-induced HUVEC cell cycle progression at the G2/M phase and is closely related to the expression of cell cycle-related protein.

### 2.6. Effect of R-PC on the Lifespan of Drosophila melanogaster and the Climbing Ability of Drosophila melanogaster

Weight changes in drosophila can be regarded as a pointer to feed intake. Although, the weight of the female experimental groups did not change remarkably compared with the control group. It showed that the weight of female flies was not decreased through R-PC (Figure 6A). However, the male experimental groups had a remarkable weight added after feeding R-PC (20 mg/mL, 25 mg/mL, 30 mg/mL) or resveratrol. In addition, female flies showed the most significant improvement in the 30 mg/mL R-PC group, with the average lifespan increasing from 50.2 days to 73.4 days and the maximum lifespan added from 59.1 days to 77.2 days, which were remarkable (Figure 6B–D). These data suggested that R-PC showed good longevity effects on female flies.

*Drosophila melanogaster* has a negative physical tendency to climb. When falling to the bottom of the bottle, young and robust drosophila climb up at once, but aging drosophila cannot climb up or fall back to the bottom of the bottle after a period of crawling. To examine the effect of R-PC on the climbing ability of natural aging drosophila, we investigated the climbing ability of flies. As shown in Figure 6E, R-PC had a clear improvement in the climbing ability of female drosophila. While feeding 25 mg/mL or 30 mg/mL of R-PC in the diets, the climbing ability of drosophila increased remarkably. However, the climbing ability of the male drosophila was not significant.

### 2.7. Effect of R-PC on ROS Levels and Antioxidant-Related Gene Expression in Drosophila melanogaster

To examine the effect of supplementing different concentrations of R-PC on the levels of ROS in natural aging drosophila, we used DCFH-DA to detect intracellular ROS to investigate the protective effect of R-PC on oxidative damage in natural aging drosophila. The ROS levels in the female control drosophila were added notably on the 28th day (*p* < 0.001) and reduced clearly after the addition of 30 mg/mL of R-PC (*p* < 0.001) (Figure 7B). The effect of R-PC on reducing the level of ROS was more obviated in the female groups. On the 28th day, R-PC tended to reduce the ROS level in female flies. On the 56th day, when the R-PC addition concentrations were 25 mg/mL and 30 mg/mL in female flies, the levels of ROS were reduced by 1.89 and 2 in female flies, respectively, and the levels of ROS were reduced by 1.59 and 1.94 in male flies, respectively (Figure 7C). Results showed the effective protection of R-PC against oxidative damage in natural aging female flies.

To examine whether the effect of R-PC on the relative expression levels of antioxidant-related genes, SOD1, SOD2, and CAT were tested by the q-PCR. Results showed that the SOD1, SOD2, and CAT gene expressions have varying degrees of increase in the R-PC group compared with the control group, and supplementation with R-PC in the diets can significantly up-regulate the expression of these genes in female flies (Figure 8A–C). These results showed that R-PC regulated antioxidant enzyme activities by increasing the expression of antioxidant-related genes in the experimental group, improving the antioxidant enzyme system in natural aging drosophila, and extending its lifespan.

## 3. Discussion

At present, anti-aging means improving the physical quality of life and extending longevity within the allowable range of genetic characteristics. Many methods have been explored, including gene therapy, nutritional modulation, hormonal supplementation, and intervention by antioxidants and other compounds [28]. The health benefits of phycocyanin or phycocyanin-rich seaweed have been broadly reported. Consumption of phycocyanin from seaweed protects against neurodegenerative diseases and age-related cognitive decline [29]. R-PC, one of the types of phycocyanin, contains open-chain tetra pyrroles with putative scavenging properties. Still, minimal information is available on the anti-aging effects of R-PC-rich *Porphyra haitanensis*. The present work complements this study.

Several studies showed that phycocyanin is a heterodimer of α (Cpcα, Cphycocyanin alpha-subunit gene product) and β (Cpcβ, phycocyanin beta-subunit gene product) subunits of the phycobiliprotein monomer [30]. The heterodimer is very stable and can be assembled into disc-like trimers (αβ)_3_, which can further stack to form a hexamer (αβ)_6_. The hexamers are rapidly collected into the phycobilisome rod substructures through interactions with the linker proteins in the form of linker–hexamer complexes. The structures caused an excitation energy transfer phenomenon between light-harvesting pigments, tryptophan residues, and chromophores, which can transform from the ground state to the excited state and can transfer electrons, including oxidative stress-mediated damage [31]. In this study, our results showed that R-PC exists as high molecular aggregates in porphyra haitanensis. The subunit organization is based on a heterodimer (αβ, conventionally called monomer) composed of a- and b-subunits with unequal molecular weights, including one α subunit (α_18.5_^6.5^), and four β subunits (β_21.2_^5.5^, β_23.9_^5.5^, β_21.2_^5.6^, β_23.9_^5.6^) with different PIs and molecular weights (Figure 1). Studies have shown that phycocyanin α and β subunits expressed in E. coli have antioxidant activity and can inhibit the generation of hydroxyl radicals. Guan found that the phycocyanin α subunit has a strong antioxidant capacity and can effectively scavenge hydroxyl and peroxy radicals [32]. Yu’s research also demonstrated that the phycocyanin alpha subunit scavenges free radicals and protects cell growth from oxidative damage by DPPH and H_2_O_2_ [33]. Studies showed the close relationship between αβ subunits and antioxidant activity, and then we tested whether antioxidant activity in R-PC.

Oxidative stress-mediated damage to functional macromolecules followed by their accumulation has been reported as a major factor behind aging and related consequences [34]. Recent studies showed that C-phycocyanin has significant radical-scavenging activity and its utilization as a natural colorant in the food industry. C-phycocyanin from Limnothrix had an antioxidative activity on DPPH free radicals similar to that found in a natural antioxidant-rutin. Romay et al. demonstrated that C-phycocyanin could scavenge alkoxyl, hydroxyl, and peroxyl radicals and react with peroxynitrite (ONOO–) and hypochlorous acid (HOCl). In addition, they showed that C-phycocyanin inhibited microsomal lipid peroxidation [35]. Similarly, in this study, ABTS assay, FRAP assay, and DPPH assay were tested to investigate the antioxidant activity of R-PC in vitro, and the scavenging activities of R-PC on free radicals exhibited an explicit concentration dependency. R-PC showed great antioxidant activity in vitro and potential use as antioxidants. Next, we investigated the relationship between antioxidants and anti-aging by the cells and drosophila models of aging.

During aging, progressive productions of ROS and/or depleted cellular defense impairs the balance between pro-oxidants and antioxidants, causing a shift in cellular redox state, which causes degeneration of cells and loss of regenerative capacity to increase, and with time, the alterations caused through them ultimately induce apoptosis. Bcl-2 family members are the sentinels in the mitochondrial apoptotic pathway, whereas the anti-apoptotic Bax proteins prevent the mitochondrial cell death pathway from occurring [36]. Activation of the mitochondrial apoptotic pathway includes loss of ΔΨm and changes in the Bcl-2/Bax ratio. The release of cytochrome C from mitochondria leads to the formation of a multimeric complex and initiates activation of Caspase-3 [37]. Once caspase-3 is in action, apoptotic cell death is inevitable. The study demonstrated that C-phycocyanin protects chondrocytes from osteoarthritis (OA) progression by reducing ROS production, caspase-3 activity, and cell apoptosis [38]. In addition, Cyclin D1 is expressed in the G2/M phase, which belongs to the highly conservative protein family. CDK1/2 is an anti-apoptotic gene that plays a central role in regulating apoptosis. The functional composite cyclin D1/CDK1/2 can induce cell cycle transition from S to G2/M phase, initiate synthesis and replication of DNA, and accelerate cell division. H_2_O_2_ produces hydroxyl radicals, which are used to verify its oxidative damage to HUVEC cells. In the H_2_O_2_-challenged assay, the supplementation of a high dose of R-PC significantly prolonged the survival time, which indicated that R-PC could markedly enhance cell viability and protect HUVEC cells from oxidative stress relative to cellular injuries. H_2_O_2_-induced ultrastructural changes such as cell shrinkage, formation of membrane blebs, loss of ΔΨm, the numbers of apoptotic cells, and G2/M cell cycle arrest are some characteristics of cells undergoing apoptosis.

Further analysis showed that R-PC significantly up-regulated Bcl-2/Bax ratio and down-regulated levels of cytochrome C, which suggests the involvement of the mitochondrial apoptosis pathway. Moreover, evidence has indicated that apoptosis involves the release of cytochrome C from mitochondria, causing caspase activation. In this study, R-PC decreased the expression of Caspase-3 and PARP, which reduced the H_2_O_2_-induced HUVEC cells’ apoptosis (Figure 4 and Figure 5). Meantime, R-PC also reversed p21, p27, Cyclin D1, and CDK1/2 expressions after stimulation with H_2_O_2_. Our findings suggest that R-PC prevents the H_2_O_2_-induced cell cycle phase arrest by regulating cycle-related protein and apoptosis through mitochondrial and Caspase-3-dependent apoptosis pathways in HUVEC cells.

Reports showed that increased mitochondrial activity and decreased ROS production can trigger an intrinsic defense program, which results in improved stress resistance and possibly lifespan extension [39]. Recent studies showed that C-phycocyanin exerts an antioxidant effect by inhibiting ROS production, which decreases apoptosis. In addition, C-phycocyanin can reverse the expression of antioxidant genes in adult female B6D2F/1 mice injected with D-galactose, increase SOD activity and reduce MDA content, and normalize mitochondria distribution [40,41]. Our study showed that the high dose of R-PC effectively prolonged the mean lifespan, the 50% survival time, and the maximum lifespan of female flies. Furthermore, we found that R-PC treatment showed good longevity effects on female flies and improved the climbing ability of natural aging female flies. SOD1, SOD2, and CAT are the most critical antioxidant enzymes in the body, which can inhibit the free radical formation and mainly maintains the dynamic balance of the internal environmental ROS, removes excessive ROS, and keeps the body relatively stable state. To further clarify the functional mechanism of R-PC in natural aging drosophila, we analyzed SOD1, SOD2, and CAT genes’ expression. SOD and CAT are the first line of defense against superoxide radicals in endogenous antioxidant defense systems. These enzymes are associated with longevity. This finding indicated that excessive accumulation of free radicals could cause organ dysfunction and aging. R-PC could exert an antioxidant effect by inhibiting ROS levels, reducing the production of lipid peroxides, and abating oxidative damage in natural aging *Drosophila melanogaster*. In addition, the lifespan of female drosophila was higher than males in the same concentration group. The effect of gender on aging may be related to the differences in sensitivity to insulin or insulin-like growth factor signaling in *Drosophila melanogaster* or differences in nutritional and energy requirements, allocation, and utilization.

## 4. Materials and Methods

### 4.1. Chemicals and Reagents

The crude extract of R-phycocyanin was prepared on a laboratory scale from *Porphyra haitanensis* (the contents of crude protein 15.78 μg/mL). HUVEC cells were purchased from ATCC. *Drosophila melanogaster* was from Engineering Research Centre of Fujian-Taiwan Special Marine Food Processing and Nutrition.

Senescence-associated β-galactosidase (SA β-gal) staining, Hoechst and PI Staining Kit, Annexin V Apoptosis Detection Kit, 2′,7′-dichlorofluorescein diacetate (DCFH-DA) were purchased from the Beyotime Institute of Biotechnology (Shanghai, China). Cells counting kit-8 (CCK-8), the Mitochondrial Membrane Potential Assay Kit with JC-1, DNA content quantitation assay, and Nuclear Protein Extraction Kit were obtained from Solarbio Science & Technology Co., Ltd. (Beijing, China). ECL was purchased from Thermo Fisher Scientific (Waltham, MA, USA). All antibodies were purchased from Santa Cruz Biotechnology, Inc. (Dallas, TX, USA).

### 4.2. Isolation and Purification of R-PC

Sephadex G-150 gel column chromatography (5 cm × 62 cm or 3.6 cm × 64 cm) was used to separate R-PC from the phycobiliprotein concentrate and eluted with 50 mmol/L PBS at a speed of 60 mL/h. The obtained R-PC fraction was concentrated and then subjected to Sephadex G-150 gel column chromatography (Vt = 3.6 cm × 64 cm, elution flow rate 30 mL/h) to further remove R-PC. The R-PC sample obtained by ion-exchange chromatography gel filtration was purified by DEAE-Sepharose FF ion-exchange column (2.6 cm × 7.6 cm). Perform linear gradient elution with 600 mL of 25 mmol/L PBS in 0–600 mmol/L NaCl solution at a flow rate of 30 mL/h. The R-PC fractions were analyzed by SDS-PAGE and denaturing-IEF [15].

### 4.3. Antioxidant Capacity In Vitro

Antioxidant capacity was measured by 2,2′-azinobis (3-ethylbenzthiazoline- 6-sulfonic acid) (ABTS) assay, ferric reducing antioxidant power (FRAP) assay, and 1,1-Diphenyl-2-picryl-hydrazyl (DPPH) assay. One gram of freeze-dried R-PC sample was mixed with 20 mL of ethanol. The mixture was subjected to agitation for 10 min and centrifuged at 660× *g* for 15 min.

The FRAP assay is presented as a novel method for assessing antioxidant power. For the FRAP assay, 0.5 mL aliquot of the extract was dissolved in 3 mL of ethanol and reacted with 0.3 mL of FRAP reagent. The decrease in FRAP absorbance was measured at 517 nm using a UV/Vis spectrophotometer.

ABTS free-radical scavenging activity is the most common method for evaluating plant antioxidant activities. For the ABTS assay, 30 μL aliquot of the extract was reacted with 3 mL of ABTS reagent. The decrease in ABTS absorbance was measured at 734 nm. Ethanol was used as blank for both assays.

Scavenging of DPPH free radical is the basis of a typical antioxidant assay. The percent radical scavenging activity of methanolic extracts was quantified by allowing the 0.1 mL extract to inhibit the constant volume of DPPH solution. The absorbance of the reaction mixture was read using a spectrophotometer at 517 nm to determine the free radical scavenging activity [42].

### 4.4. Cell Cultures

Human umbilical vein endothelial (HUVEC) cells were grown in Roswell Park memorial institute 1640 (RPMI 1640; Gibco, Waltham, MA, USA). RPMI 1640 medium supplemented with 10% fetal bovine serum, 100 U/mL penicillin, and 100 μg/mL streptomycin was incubated at 37 °C enriched with 5% CO_2_.

HUVEC cells were seeded in 96-well plates at a density of 1.0 × 10^5^ cells/well and grew to 80% confluence. The cells were pretreated with resveratrol or R-PC for 120 min and then exposed to 100 µM H_2_O_2_ for another 30 min [15].

### 4.5. CCK-8 Assay

Cells were cultured for 24 h at 37 °C on 96-well plates (6 × 10^3^ cells/well) with 5% CO_2_. Then, the cells were treated with different concentrations of resveratrol or R-PC for 24 h. The cells in the control group were treated with RPMI 1640. Next, the cells were incubated with 10 μM CCK-8 at 37 °C enriched with 5% CO_2_ for 2 h. Absorption intensity was analyzed using the Bio-Tek EXL808 microplate reader (BioTek Instruments Inc., Winooski, VT, USA) at the wavelength of 450 nm. The 50% inhibitory concentration (IC50) values were calculated by GraphPad Prism 5.01 (GraphPad Software, Inc., San Diego, CA USA).

HUVEC cells were seeded in 96-well plates at a density of 1.0 × 10^5^ cells/well and grew to 80% confluence. The cells were pretreated with resveratrol or R-PC (3, 10, 20, 40 µM) for 24 h, and then exposed to 100 µM H_2_O_2_ for another 30 min.

### 4.6. SA β-Gal Staining

Cells were washed twice with PBS and fixed in 10% formaldehyde for 15 min at room temperature. Then, the cells were washed twice with PBS and incubated at 37 °C (without CO_2_) with fresh senescence-associated β-galactosidase staining solution. The color developed in 12 h. Photographs of the light microscopy images of the cells were obtained. β-galactosidase was found to be positively expressed in the cytoplasm by green staining.

### 4.7. Hoechst and PI Staining Kit

Hoechst 33324 staining was used to detect H2O2-induced HUVEC cells apoptosis. HUVEC cells seeded in 6-well plates (1 × 10^5^ cells/well) were then treated with resveratrol or R-PC. Next, cells were stained with 5 μL Hoechst 33324 and 10 μL propidium iodide (PI; Beyotime Institute of Biotechnology, Shanghai, China) stain for 5 min at 37 °C, then observed under a fluorescence microscope DM 2500 (Leica Microsystems GmbH, Wetzlar, Germany) at magnification, ×400.

### 4.8. Annexin V-Fluorescein Isothiocyanate (FITC)/PI Double Staining and Flow Cytometry

H_2_O_2_-induced HUVEC cells were seeded in 6-well plates (1 × 10^5^ cells/well) and were then treated with resveratrol or R-PC. Next, cells were stained with 2 μL Annexin V-FITC and 10 μL PI. Samples were measured using flow cytometry (Beckman Coulter, Brea, CA, USA). Finally, the cell apoptosis number was analyzed by CytExpert software 2.0 (Beckman Coulter, Inc., Brea, CA, USA).

### 4.9. Mitochondrial Membrane Potential Analysis

H_2_O_2_-induced HUVEC cells were cultured in 6-well plates (1 × 10^5^ cells/well) and with R-PC. Subsequently, cells were treated with JC-1 (10 μg/mL) at room temperature for 5 min to detect mitochondrial membrane potential (ΔΨm) depolarization. Flow cytometry determined states of ΔΨm depolarization, and CytExpert software was used to analyze the data. Data are given as the relative ratio of green to red fluorescence intensity, showing the level of depolarization of the mitochondrial membrane potential.

### 4.10. Cell Cycle Analysis

H_2_O_2_-induced HUVEC cells were cultured in 6-well plates (1 × 10^5^ cells/well) and treated with R-PC. Subsequently, the cells were fixed with 70% ethanol for 12 h at −4 °C. Next, cell suspensions were incubated with RNase A (Beyotime Institute of Biotechnology, Shanghai, China) and PI for 30 min at 37 °C in the dark. The cells were then measured for DNA content by flow cytometry. The cell cycle was analyzed by CytExpert software 2.0.

### 4.11. Measurement of Intracellular ROS Levels and Flow Cytometry

Flies were raised on various diets for 7 and 21 days, sacrificed by CO2 anesthesia, and then frozen at −20 °C. Then, the mixture of 100 mg of flies and 0.9 mL of normal saline was homogenized in an ice bath and centrifuged at 10,000 rpm for 10 min to prepare 10% tissue homogenate. The levels of ROS were measured by DCFH-DA and flow cytometry. CytExpert software was used to analyze the data.

### 4.12. Survival and Life Span Studies

In order to examine the effects of R-PC on the lifespan of *Drosophila melanogaster*, the drosophila melanogaster was randomly allocated to the control or experimental groups. *Drosophila melanogaster* was bred and maintained on a simple diet prepared using 100 g corn flour, 20 g baker’s yeast, 12 g Agar-agar, 1 g of methylparaben in 5 mL ethanol, and 1700 mL water. Fly laboratory temperature was kept constant at 25 °C. Living *Drosophila melanogaster* were transferred to a new pipe containing the same medium every three days. Thirty synchronized flies were placed in each vial, and each dosage group had 10–15 vials. The number of surviving *Drosophila melanogaster* was counted daily, and the culture medium was changed every 4 days until all flies died. The average life expectancy of all drosophila and the time until the death of 90% of drosophila in a given treatment group were considered to be the mean and maximum life span, respectively. The lifespan assays were repeated three times. Half days of death were regarded as the average lifespan of the 50% surviving *Drosophila melanogaster*.

### 4.13. Climbing Ability Assays

Climbing assays were used to evaluate the locomotor function of *Drosophila melanogaster*. In short, wild-type flies were divided into control or R-PC groups. Ten flies were shaken into the bottom of the empty plastic vials and given the 20 s to climb up. In each trial, we recorded the number of flies that climbed vertically to 5 cm and above [18].

### 4.14. Western Blot Assay

H_2_O_2_-induced HUVEC cells were treated with R-PC, extracted with a cell lysis buffer (1 M HEPES, pH 7.0; 5 M NaCl; 0.5% Triton X-100; 10% glycerol; 20 mM β-mercaptoethanol; 20 mg/mL AEBSF; 0.5 mg/mL pepstatin; 0.5 mg/mL leupeptin; and 2 mg/mL aprotinin) for 30 min and centrifuged at 12,000× *g* for 30 min at 4 °C, and total protein was quantified using coomassie blue staining. Equivalent proteins (30 μg) were separated by 8–12% sodium dodecyl sulfate-polyacrylamide gel electrophoresis and transferred onto nitrocellulose membranes, which were incubated in blocking solution (fresh 5% non-fat milk in 10 mM Tris-HCl containing 150 mM NaCl (TBS; pH 7.5) and TBS + 0.2% Tween-20 (TBST) for 1 h at room temperature. The membranes were incubated for 12 h at 4 °C with the following primary antibodies (all from Santa Cruz Biotechnology, Dallas, TX, USA), mouse monoclonal antibodies: Mouse monoclonal antibodies against and rabbit polyclonal antibodies against. The membranes were incubated with horseradish peroxidase-conjugated anti-mouse (1:5000; cat. no. ZB-2301) and anti-rabbit immunoglobulin G (1:5000; cat. no. ZB-2305) secondary antibodies for 2–3 h at room temperature, then followed by washing with TBST. Proteins were visualized using Pierce ECL Western blot substrate (Thermo Fisher Scientific, Waltham, MA, USA) and the AI600 chemiluminescence imager (GE Healthcare, Fairfield, CT, USA). They were semi-quantified using Image J version 1.46r (National Institutes of Health, Bethesda, MD, USA).

### 4.15. Quantitative Polymerase Chain Reaction (q-PCR) Assay

*Drosophila melanogaster* was fed with the corresponding diets for 56 days. The total RNA of flies was extracted, and then q-PCR was carried out on a Bio-Rad CFX Connect™ Real-Time System (Bio-Rad Laboratories, Inc., Hercules, CA, USA).

The antioxidant genes included SOD1 (5′-GCGGCGTTATTGGCATTG-3′, 3′-ACTAACAGACCACAGGCTATG-5′), SOD2 (5′-CACATCAACCACACCATCTTC-3′, 3′-GCTCTTCCACTGC GACTC-5′) and CAT (5′-TGAACTTCCTGGATGAGATGTC-3′, 3′-TCTTGGCGGCACAATACTG-5′), which were normalized with RP49 (5′-AGGGTATCGACAACAGAGTG-3′, 3′-CACCAGGAACTTCTTGAATC-5′), a housekeeping gene used for normalization. The expression of the target genes was calculated using the 2−ΔΔCt method [43,44].

### 4.16. Statistical Analysis

Data were evaluated by the analysis of variance, and differences between groups were evaluated by one analysis of variance. Tukey’s post hoc tests were performed using the Statistical Package for Social Science (SPSS; version 21.0; SPSS Inc, Chicago, IL, USA). P values less than 0.05 were considered to indicate a statistically significant difference.

## 5. Conclusions

In this study, R-PC from the *Porphyra haitanensis* extract is composed of one α subunit and four β subunits (β_21.2_^5.5^, β_23.9_^5.5^, β_21.2_^5.6^, β_23.9_^5.6^) with different PIs. The structures had a tight link with antioxidation, and R-PC revealed interested antioxidant effects in this study. Action mechanism studies showed that the anti-aging effects of R-PC are mainly due to their antioxidative ability. R-PC can have a significant protective effect against H_2_O_2_-induced HUVEC cell death. R-PC exerts an antioxidant effect by removing excessive ROS and activating the antioxidant enzymes (CAT, SOD1, and SOD2) to protect *Drosophila melanogaster* against oxidative stress injury and keep the dynamic balance of the internal environmental ROS, finally improving impairments in natural aging drosophila. R-PC is a hopeful anti-aging functional food with protective activities against aging-caused oxidative stress.

## Figures and Tables

**Figure 1 marinedrugs-20-00468-f001:**
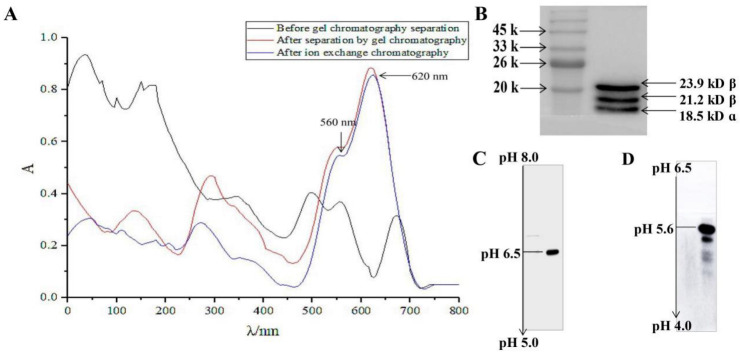
Isolation and purification of R-PC. (**A**) The absorption spectra of the porphyra haitanensis extract, the RPC fractions from the Sephadex G-150 gel filtration and the DEAE-Sepharose FF chromatography. (**B**) The gradient SDS-PAGE of the purified RPC. (**C**) The denaturing-IEF of α subunit of the purified R-PC. (**D**) The native-IEF of the R-PC purified by the ion-exchange chromatography.

**Figure 2 marinedrugs-20-00468-f002:**
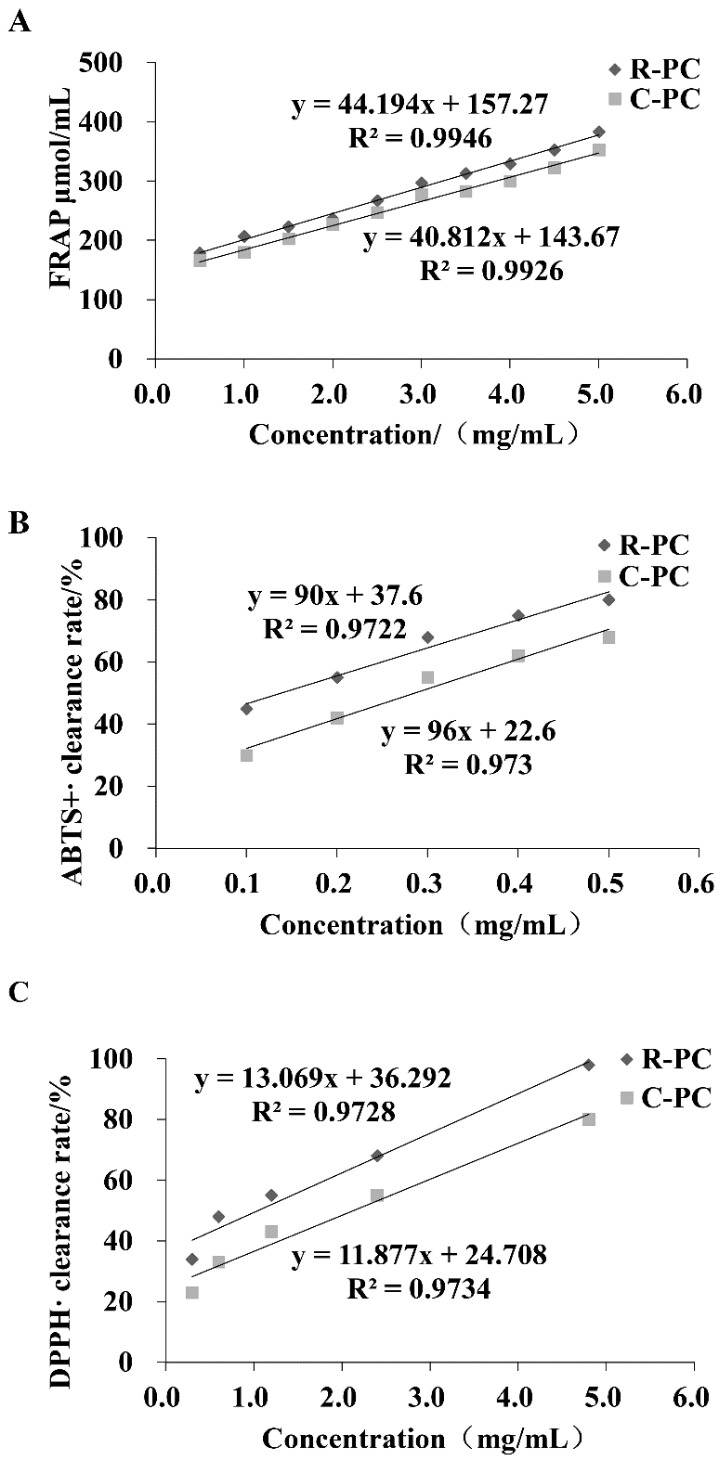
Antioxidant capacity in vitro. (**A**) Fe^3+^ reducing ability of different phycocyanins in FRAP system. (**B**) The scavenging ability of different phycocyanins against ABTS^+^. (**C**) The scavenging ability of different phycocyanins against DPPH.

**Figure 3 marinedrugs-20-00468-f003:**
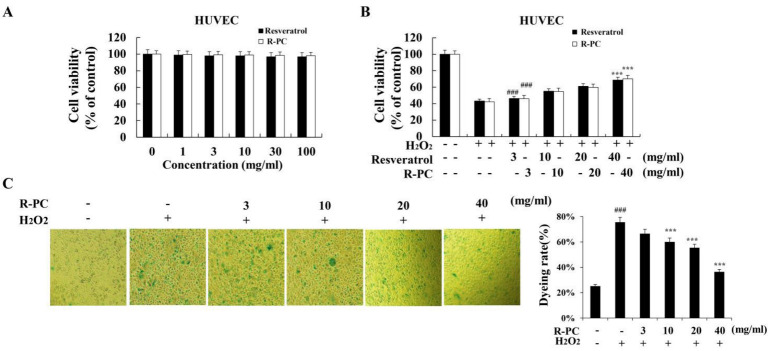
R-PC inhibits H_2_O_2_-induced cytotoxicity in HUVEC cells. (**A**) HUVEC cells’ viabilities were assessed using the CCK-8 assay. Cells were incubated with resveratrol or R-PC at the indicated concentration for 24 h intervals, followed by incubation with CCK-8 solution for 2 h. (**B**) HUVEC cells pretreated with resveratrol or R-PC (3, 10, 20, 40 mg/mL) for 24 h were exposed to 100 μM H_2_O_2_ for 30 min. (**C**) Effects of age on senescence-associated β-galactosidase staining. ^###^ *p* < 0.001 versus untreated group (control); *** *p* < 0.001 versus H_2_O_2_ alone group.

**Figure 4 marinedrugs-20-00468-f004:**
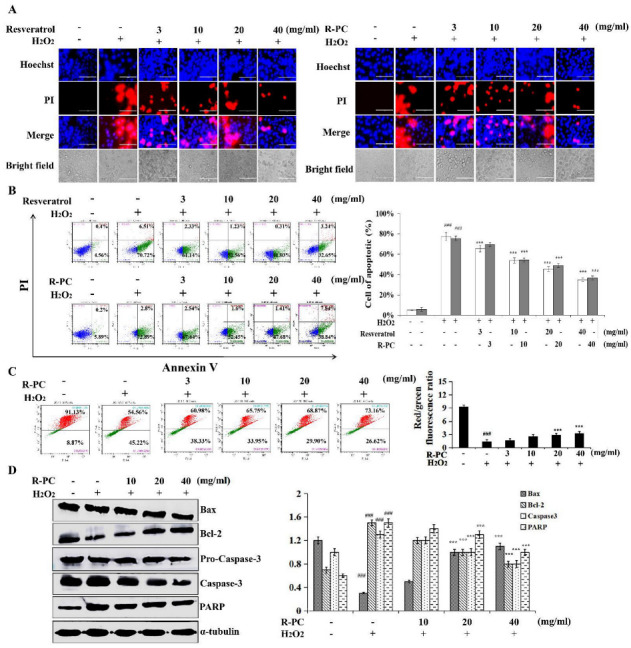
P-RC protects against H_2_O_2_-induced apoptotic cell death in HUVEC cells. (**A**) HUVEC cells pretreated with resveratrol or R-PC (3, 10, 20, 40 mg/mL) for 24 h were exposed to 100 μM H_2_O_2_ for 30 min, and observed by fluorescence microscopy. Scale bar = 200 µm. (**B**) HUVEC cells pretreated with resveratrol or R-PC (3, 10, 20, 40 mg/mL) for 24 h were exposed to 100 μM H_2_O_2_ for 30 min, and apoptosis was analyzed by flow cytometry. (**C**) HUVEC cells pretreated with resveratrol or R-PC (3, 10, 20, 40 mg/mL) for 24 h were exposed to 100 μM H_2_O_2_ for 30 min. ΔΨm was measured by JC-1 and analyzed by flow cytometry. (**D**) The expression levels of Bax, Bcl-2, pro-caspase-3, Caspase-3 and PARP proteins were analyzed by western blot. Error bars indicate means ± SDs of three independent experiments. ^###^ *p* < 0.001 versus untreated group (control); *** *p* < 0.001 versus H_2_O_2_ alone group.

**Figure 5 marinedrugs-20-00468-f005:**
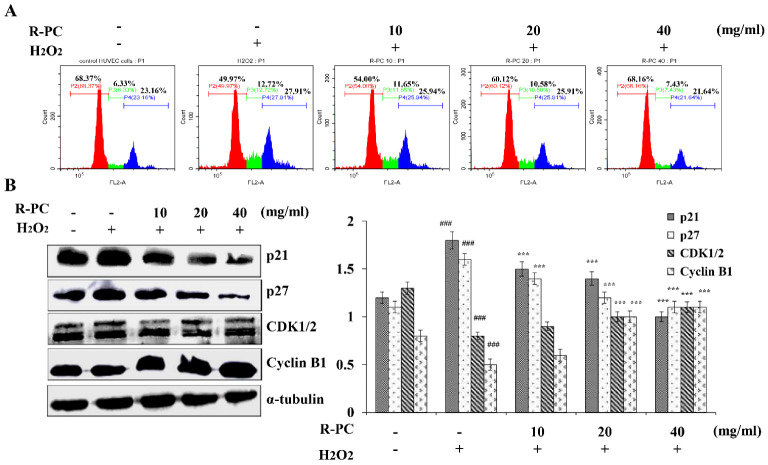
R-PC prevents H_2_O_2_-induced cell cycle progression at the G0/G1 phase in HUVEC cells. (**A**) HUVEC cells pretreated with resveratrol or R-PC (10, 20, 40 mg/mL) for 24 h were exposed to 100 μM H_2_O_2_ for 30 min, and effects on the cell cycle were analyzed by flow cytometry. (**B**) The expression levels p21, p27, cyclin B1 and CDK1/2 proteins were analyzed by western blot. Error bars indicate means ± SDs of three independent experiments. ^###^ *p* < 0.001 versus untreated group (control); *** *p* < 0.001 versus H_2_O_2_ alone group.

**Figure 6 marinedrugs-20-00468-f006:**
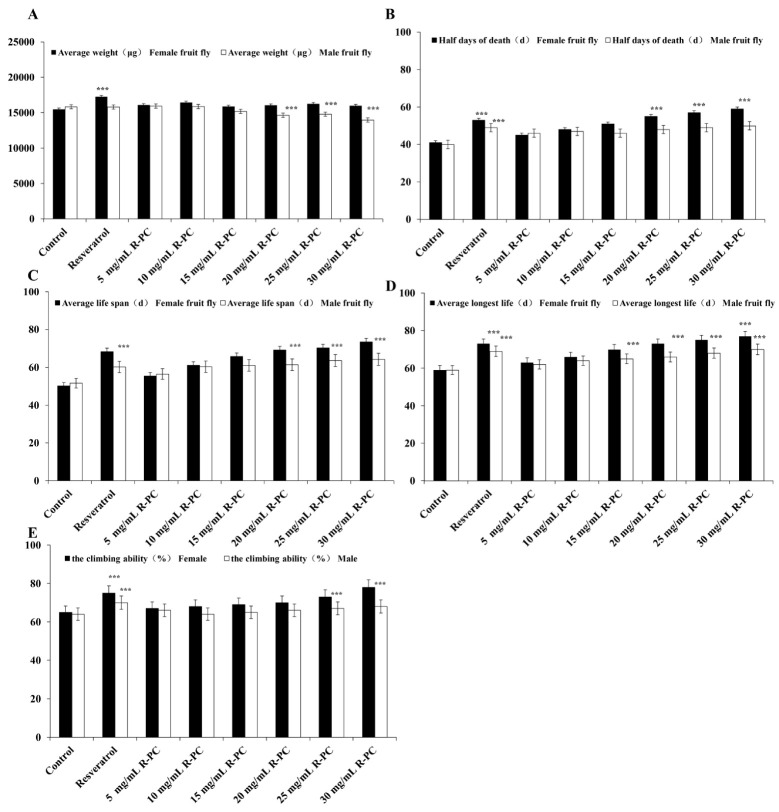
Effect of R-PC on the lifespan of *Drosophila melanogaster* and the climbing ability of drosophila melanogaster. (**A**) Effects of different diets on average body weight of drosophila. (**B**) Half days of the death of drosophila reared on different diets. (**C**) Average lifespan of drosophila reared on different diets. (**D**) Maximum lifespan of drosophila reared on different diets. Male and female drosophila (200 flies per group, 25 flies per vial). Control, control group; Resveratrol, resveratrol control group. R-PC, containing 5 mg/mL, 10 mg/mL, 15 mg/mL, 20 mg/mL, 25 mg/mL, 30 mg/mL of R-PC, respectively. (**E**) Male and female drosophila (200 flies per group, 25 flies per vial). Control, control group; Resveratrol, resveratrol control group. R-PC, containing 5 mg/mL, 10 mg/mL, 15 mg/mL, 20 mg/mL, 25 mg/mL, 30 mg/mL of R-PC, respectively. Error bars indicate means ± SDs of three independent experiments. *** *p* < 0.001 versus control group.

**Figure 7 marinedrugs-20-00468-f007:**
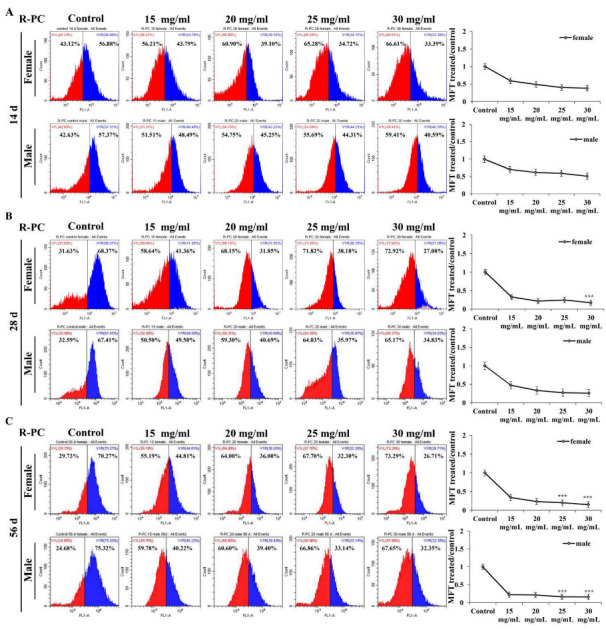
Effect of R-PC on ROS levels and antioxidant-related gene expression in *Drosophila melanogaster*. (**A**) *Drosophila melanogaster* were treated with R-PC for 14 days, and the intracellular ROS levels were analyzed by flow cytometry. (**B**) *Drosophila melanogaster* were treated with R-PC for 28 days, and the intracellular ROS levels were analyzed by flow cytometry. (**C**) *Drosophila melanogaster* were treated with R-PC for 56 days, and the intracellular ROS levels were analyzed by flow cytometry. Error bars indicate means ± SDs of three independent experiments. *** *p* < 0.001 versus control group.

**Figure 8 marinedrugs-20-00468-f008:**
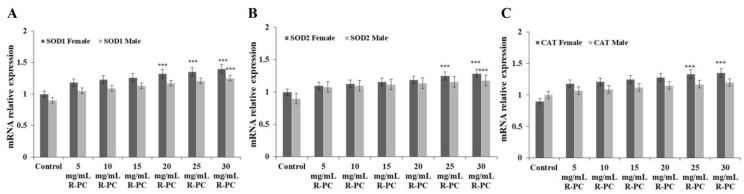
Effect of R-PC on ROS levels and antioxidant-related gene expression in *Drosophila melanogaster*. (**A**) *Drosophila melanogaster* were treated with R-PC for 56 days, the expression levels of SOD1 were analyzed by q-PCR. (**B**) The expression levels of SOD2 were analyzed by q-PCR. (**C**) The expression levels of CAT were analyzed by q-PCR. Error bars indicate means ± SDs of three independent experiments. *** *p* < 0.001 versus control group.

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
