# Peer review of "Anti-Aging Effects of R-Phycocyanin from Porphyra haitanensis on HUVEC Cells and Drosophila melanogaster"

_marinedrugs, 2022, doi:10.3390/md20080468_

Round 1
Reviewer 1 Report
The manuscript entitled 'Anti-aging effect of R-Phycocyanin from Phorphyra haitanensis on HUVEC cells and Drosophila melanogaster' by Yanyu Feng et al. investigates the effect of R-Phycocyanin contained in the algae Phorphyra haitanensis as an anti-oxidant and anti-aging agent.
The experimental model is exhaustive and adequate as it evaluates the effect of R-Phycocyanin on both cultured cells "in vitro" and animal models "in vivo". The study is well defined and experiments have been appropriately conducted. The results are also interesting from a commercial point view.
Minor concerns are as follows:
In the introduction there are syntax errors in line 31 and in the sentence line 38-41.
In figure 2 put FRAP instead of FRAR in the graph
In the caption of figure 2 insert the meaning of Clearance rate%.
There is an error in line 155
In Figure 4D the pro-caspase spots appear to be drawn. The authors are requested to provide the original membrane to verify the authenticity of the western blotting performed
In figure 5D the spots are unclear and therefore difficult to quantify. In this case the authors are requested to replicate the western
In section 2.6 line 251 the authors say that there is weight gain in males but this does not appear to be true from the graphs shown in figure 6.
The graphs in figure 7 D-E-F I would move them to an additional figure because they are very small and not easy to read
In line 364 what does OA mean?
Author Response
Dear Editor:
Thank you for your letter and for the reviewers’ comments concerning our manuscript entitled “Anti-aging effects of R-phycocyanin from porphyra haitanensis on HUVEC cells and drosophila melanogaster” (ID: marinedrugs-1807399). Those comments are all valuable and very helpful for revising and improving our paper, as well as the important guiding significance to our researches. We have studied comments carefully and have made correction which we hope meet with approval. Improved portion are marked in red in the revised manuscripts. Because the most current version has added some words, this caused the line numbers has been changed. The main corrections in the paper and the responds to editor’s and review’s comments are in following next page.
Thank you and best regards.
Sincerely,
Yi Zhang, Ph.D. Professor:
Fujian Agriculture and Forestry University, Fuzhou, Fujian, 350000, China.
E-mail: [email protected]
Responds to the reviewer’s comments:
Reviewer #1:
- Comment 1: In the introduction there are syntax errors in line 31 and in the sentence line 38-41.
- Response 1: Thanks for reviewer’s kind suggestion. According to reviewer’s comment, we have edited and corrected all typos in manuscript by language editing and native. The inserted explanations have been added in the revised manuscript as follow:
Page 1, Line 31~:
At present, numerous older people will devote themselves to keeping a healthy body. However, this segment of the population is subject to physical and cognitive impairment at higher rates than younger people [1,2]. Older people can improve their quality of life by supplying anti-aging ingredients in their diet.
Page 1, Line 38~41:
Oxidative stress occurs due to the relatively imbalanced state between production and elimination of ROS in the body. It causes the decline of the body’s antioxidant stress system and various aging-related diseases, such as atherosclerosis, hypertension, arthritis, neurodegenerative disorders [5].
- Comment 2: In figure 2 put FRAP instead of FRAR in the graph.
- Response 2: Thanks for reviewer’s kind suggestion. Those comments are all valuable and very helpful for revising and improving our paper, as well as the important guiding significance to our researches. The relevant illustrations were added in revised Figure 2 as follow:
Revised Figure 2:
- Comment 3: In the caption of figure 2 insert the meaning of Clearance rate %.
- Response 3: Thanks for reviewer’s kind suggestion. Those comments are all valuable and very helpful for revising and improving our paper, as well as the important guiding significance to our researches. According to reviewer’s comment, we have added the meaning of clearance rate to the caption of figure 2 as follow:
Revised Figure 2:
Page 4, Line 134:
Figure 2. Antioxidant capacity in vitro. (A) Fe3+ reducing ability of different phycocyanins in FRAP system. (B) The scavenging ability of different phycocyanins against ABTS+ž. (C) The scavenging ability of different phycocyanins against DPPHž.
- Comment 4: There is an error in line 155.
- Response 4: Thanks for reviewer’s kind suggestion. According to reviewer’s comment, we have edited and corrected all typos in manuscript by language editing and native. The inserted explanations have been added in the revised manuscript as follow:
Page 4, Line 155 ~:
In addition, SA-β-gal is commonly regarded as an aging cell positive indicator, so this study detected whether R-PC protects cells aged after H2O2 treatment by SA-β-gal staining (figure 3B). After treatment with H2O2, the number of positive cells with blue particles increased, while R-PC protected cells from H2O2-induced cells aged in a con-centration-dependent manner (Figure. 3C).
- Comment 5: In Figure 4D the pro-caspase spots appear to be drawn. The authors are requested to provide the original membrane to verify the authenticity of the western blotting performed.
- Response 5: Thanks for reviewer’s kind suggestion. Those comments are all valuable and very helpful for revising and improving our paper, as well as the important guiding significance to our researches. According to reviewer’s comment, we have provided the original membrane in appendix to verify the authenticity of the western blotting performed and added in revised Figure 4 as follow:
Revised Figure 4:
The original membrane:
- Comment 6: In figure 5D the spots are unclear and therefore difficult to quantify. In this case the authors are requested to replicate the western.
- Response 6: Thanks for reviewer’s kind suggestion. Those comments are all valuable and very helpful for revising and improving our paper, as well as the important guiding significance to our researches. According to reviewer’s comment, we have replaced the band of p21, p27, CDK1/2, cyclinB1 proteins and revised all the Figures of the manuscript to present higher-quality western blot images. The relevant illustrations were added in revised figure 5D as follow:
Revised Figure5:
- Comment 7: In section 2.6 line 251 the authors say that there is weight gain in males but this does not appear to be true from the graphs shown in figure 6.
- Response 7: Thanks for reviewer’s kind suggestion. Those comments are all valuable and very helpful for revising and improving our paper, as well as the important guiding significance to our researches. Weight changes in drosophila can be regarded as a pointer of feed intake. The weight of the female experimental groups did not change remarkably compared with the control group, it showed that the weight of female flies was not decreased through R-PC. However, the male experimental groups had a remarkably weight added after feeding R-PC (20 mg/ml, 25 mg/ml, 30 mg/ml) or resveratrol. According to reviewer’s comment, the inserted explanations have been added in the revised manuscript as follow:
Page 7, Line 248:
Weight changes in drosophila can be regarded as a pointer to feed intake. Although, the weight of the female experimental groups did not change remarkably compared with the control group. It showed that the weight of female flies was not decreased through R-PC (Figure. 6A). However, the male experimental groups had a remarkable weight added after feeding R-PC (20 mg/mL, 25 mg/mL, 30 mg/mL) or resveratrol. In addition, female flies showed the most significant improvement in the 30 mg/mL R-PC group, with the average lifespan increasing from 50.2 days to 73.4 days and the maximum lifespan added from 59.1 days to 77.2 days, of which were remarkable (Figure. 6B-D). These data suggested that R-PC showed good longevity effects on female flies.
- Comment 8: The graphs in figure 7 D-E-F I would move them to an additional figure because they are very small and not easy to read.
- Response 8: Thanks for reviewer’s kind suggestion. Those comments are all valuable and very helpful for revising and improving our paper, as well as the important guiding significance to our researches. According to reviewer’s comment, we have moved figure 7 D-E-F to an additional figure as follow:
Revised Figure7:
Figure 7. Effect of R-PC on ROS levels and antioxidant-related gene expression in Drosophila mela-nogaster. (A) Drosophila melanogaster were treated with R-PC for 14 d, and the intracellular ROS levels were analyzed by flow cytometry. (B) Drosophila melanogaster were treated with R-PC for 28 d, and the intracellular ROS levels were analyzed by flow cytometry. (C) Drosophila melanogaster were treated with R-PC for 56 d, and the intracellular ROS levels were analyzed by flow cytometry. Error bars indicate means ± SDs of three independent experiments. ***p<0.001 versus control group.
Revised Figure8:
Figure 8. Effect of R-PC on ROS levels and antioxidant-related gene expression in Drosophila mela-nogaster. (A) Drosophila melanogaster were treated with R-PC for 56 d, the expression levels of SOD1 were analyzed by q-PCR. (B) The expression levels of SOD2 were analyzed by q-PCR. (C) The ex-pression levels of CAT were analyzed by q-PCR. Error bars indicate means ± SDs of three inde-pendent experiments. ***p<0.001 versus control group.
- Comment 9: In line 364 what does OA mean?
- Response 9: Thanks for reviewer’s kind suggestion. Those comments are all valuable and very helpful for revising and improving our paper, as well as the important guiding significance to our researches. OA means osteoarthritis and it is the single most common cause of disability in older adults. Study demonstrated that C-phycocyanin protects chondrocytes from signs of osteoarthritis (OA) progression by reducing ROS production, caspase-3 activity, and cell apoptosis. The inserted explanations have been added in the revised manuscript as follow:
Page 11, Line 364:
Once caspase-3 is in action, apoptotic cell death is inevitable. The study demonstrated that C-phycocyanin protects chondrocytes from osteoarthritis (OA) progression by reducing ROS production, caspase-3 activity, and cell apoptosis [41].

Reviewer 2 Report
The article written by Feng and al. focusses on the anti-aging effects of R-Phycocyanin from Porphyra haitanensis on HUVEC cells and Drosophila melanogaster. The manuscript is very poorly written and extremely hard to understand. I cannot even understand what “H2O2-induced HUVEC” means. Figures are of poor quality also. The input of the results presented in this manuscript is very low for the topic because this question has been already explored in several papers (10.1007/s000110050256 for example
Major points
Lots of sentences are hard to understand
Figure 1 : 2 of the 3 colors are very similar. Figure 1D caption is missing
Figure 2 : axis are not readable.
Figure 3 : panel C needs a quantification
Figure 4 : too small. How the results for 20 mg/mL Resveratrol and R-PC could be explained
Figure 5 : the quality of the western blot in panel B is of extremely bad quality and cannot be quantified
Figures 6 and 7 : names on the axis are too small
Author Response
Dear Editor:
Thank you for your letter and for the reviewers’ comments concerning our manuscript entitled “Anti-aging effects of R-phycocyanin from porphyra haitanensis on HUVEC cells and drosophila melanogaster” (ID: marinedrugs-1807399). Those comments are all valuable and very helpful for revising and improving our paper, as well as the important guiding significance to our researches. We have studied comments carefully and have made correction which we hope meet with approval. Improved portion are marked in red in the revised manuscripts. Because the most current version has added some words, this caused the line numbers has been changed. The main corrections in the paper and the responds to editor’s and review’s comments are in following next page.
Thank you and best regards.
Sincerely,
Yi Zhang, Ph.D. Professor:
Fujian Agriculture and Forestry University, Fuzhou, Fujian, 350000, China.
E-mail: [email protected]
Responds to the reviewer’s comments:
Reviewer #2:
- Comment 1: Lots of sentences are hard to understand.
- Response 1: Response 1: Thanks for reviewer’s kind suggestion. According to reviewer’s comment, we have edited and corrected all typos in manuscript by language editing and native. The inserted explanations have been added in the revised manuscript as follow:
Page 1, Line 27~:
The world is in the midst of an aging population challenge, which means moving from a world in which the majority of the population is relatively young to one in which a significant proportion of people are over the age of 65. At present, numerous older people will devote themselves to keeping a healthy body. However, this segment of the population is subject to physical and cognitive impairment at higher rates than younger people [1,2]. Older people can improve their quality of life by supplying anti-aging in-gredients in their diet.
Page 1, Line 33:
A gradual decline in functional organ reserves is characterized by aging. It decreased the ability to maintain homeostasis under conditions of stress. The accrual of oxidative stress and DNA damage has been the predominant mechanistic explanation for the process of aging for many years [3]. Aging induces changes in the morphology and functions of the tissues and organs of the body change with age, which is closely related to oxidative stress [4]. Oxidative stress occurs due to the relatively imbalanced state between production and elimination of ROS in the body. It causes the decline of the body’s antioxidant stress system and various aging-related diseases, such as atherosclerosis, hypertension, arthritis, neurodegenerative disorders [5]. Mitochondria are major deter-minants of cell life and apoptosis. In normal physiological conditions, people’s bodies have internal antioxidant enzyme defense systems, such as SOD, CAT and GSH-Px, and the production and cleaning of free radicals are in a dynamic balance [6,7].However, this balance is broken when the external environment stimulates the dynamic balance, and the free radicals accumulate in large quantities, leading to mitochondria injury, which further causes cell death and metabolism disorders, causing damage to cells and tissues [8]. Antioxidants can slow the damage caused by oxidative stress and regulate numerous signaling pathways that control a wide range of cellular processes, including cell cycle and apoptosis. Thus, supplementation with natural antioxidants in diets is a relatively easy and valid way to dominate oxidative stress.
Page 2, Line 53~:
In China, Porphyra haitanensis is a sort endemic and the first to be artificially culti-vated. It has had the most considerable biomass output among many Porphyra crops in China, Japan and South Korea [9]. Recently, many natural compounds from Porphyra haitanensis have shown excellent activity against aging in vitro and in vivo. Phycocyanin is a light harvesting protein-pigment complex affiliated to the phycobilisome cores and a significant constituent of Porphyra haitanensis. Phycocyanins are divided into sub-categories according to their sources and spectral characteristics, supplemented by the prefixes R-, B-, C-, etc. Currently, C-phycocyanin has significant radical-scavenging activity, in addition to its utilization as a natural colorant in the food industry. R-phycocyanin (R-PC) is a biliprotein pigment with an open-chain tetra pyrrole chro-mophore and has various critical biological properties [10,11]. The study reported that R-PC from Porphyra haitanensis has excellent antioxidant and radical-scavenging charac-teristics, thereby protecting against antioxidative stress.
Page 2, Line 67~:
In recent years, oxidative stress has been one of the most critical mechanisms of cellular senescence, and H2O2 has been the most commonly used inducer for stress-induced premature senescence (SIPS) in cells, which shares features with replicative senescence [12,13]. At present, H2O2-induced cell models have identified several natural anti-aging active substances, such as C-phycocyanin and resveratrol [14,15]. Meanwhile, the molecular pathways that determine the aging process and prolong life have been deeply studied using animal models, and significant progress has been made. However, it is essential that data obtained from a mouse model may not extrapolate directly to natural aging, and more comprehensive research is needed in a natural aging model [16]. Drosophila melanogaster has a short lifespan, fast reproduction, easy feeding conditions, and similar metabolic pathways and aging genes to people’s bodies [17,18]. This study aimed to investigate the link between R-PC structures and anti-oxidation, and the potential anti-aging effects of R-PC were investigated on cell apoptosis, cycle, Drosophila melanogaster lifespan, level of ROS and critical molecular proteins.
Page 2, Line 83~:
As shown in Figure 1A, the phycobili protein extract was separated by Sephadex G-150 gel chromatography with the R-PC absorbance at 280 nm, 499 nm, 545 nm and 615 nm. The characteristic absorption peaks of phycoerythrin decreased significantly, while the characteristic absorptions of R-PC increased significantly at 545 nm and 615 nm. The result indicated that a large amount of phycoerythrin and some leucoproteins, which were absorbed at 280 nm had been removed by Sephadex G-150 gel chromatography. However, a weak phycoerythrin characteristic absorption can still be observed at 499 nm, and the ratio of A615/A499 was about 3, indicating that the obtained R-PC sample still contained a small amount of RPE, which needed further purification. After purification by DEAE-Sepharose FF chromatography, the obtained R-PC sample only showed the characteristic absorption peaks of R-PC at 545 nm and 615 nm, and the ratio of A615/A499 increased to 8, indicating that R-PC was further purified.
Page 4, Line 141~:
To assess whether the cytotoxic activity of R-PC against HUVEC cells, cells were respectively incubated with 0, 1, 3, 10, 30 and 100 mg/mL resveratrol or R-PC, after which cell viability was investigated by the CCK-8 assay. R-PC is not cytotoxic since doses up to 100 mg/mL did not alter the cell viability in HUVEC cells (figure 3A). The study showed that L02 cells were damaged by H2O2, with cell viability of only 40%. However, the C-phycocyanin groups restored cell viability. The more the C-phycocyanin concentration was, the better the restoration became. MDCK cells treatment with 20 mmol/L C-phycocyanin remarkably enhanced the viability of MDCK cells following 24 and 48 h treatment to oxalate [25,26]. In this study, to investigate whether R-PC has protection against oxidative stress, HUVEC cells were pretreated with resveratrol or R-PC for 24 h, and then incubated by H2O2 for 30 min, and cell viability was explored through CCK-8 assay. The viability of HUVEC cells was reduced markedly through 100 µM H2O2 for 30 min, while R-PC protected cells from H2O2-induced cytotoxicity in a concentration-dependent manner (10 mg/mL, 54.6±4.1%; 20 mg/mL, 59.8±5.0%; 40 mg/mL, 71.2±7.6%), as shown in figure 3B. R-PC could markedly enhance the cell viability at 40 mg/mL (p<0.001 versus the H2O2 alone group). In addition, SA-β-gal is a commonly used biomarker for aging cells, so we detected whether R-PC protects cells aged after H2O2 treatment by SA-β-gal staining. In addition, SA-β-gal is commonly regarded as an aging cell positive indicator, so this study detected whether R-PC protects cells aged after H2O2 treatment by SA-β-gal staining (figure 3B). After treatment with H2O2, the number of positive cells with blue particles increased, while R-PC protected cells from H2O2-induced cells aged in a concentration-dependent manner (Figure. 3C). These results showed that R-PC could protect HUVEC cells from oxidative stress relative to cellular injuries.
Page 5, Line 171~:
To investigate whether R-PC decreases the apoptosis of HUVEC cells caused through H2O2, cells were treated with R-PC, and Hoechst/PI staining, flow cytometry, JC-1 assay and western blot were used to detect apoptosis. A microscopic fluorescence assay revealed the uniform amorphous of normal HUVEC cell’s nucleus and well-distributed deep blue fluorescence. H2O2-induced apoptosis cells showed typical apoptotic morphology, including broken membranes exhibiting nuclear red fluorescence in cells (Figure. 4A). Resveratrol, a natural compound extracted from the skins of grapes, berries, or other fruits, has been shown to have anti-aging, anti-apoptotic and anti-oxidative effects. R-PC also revealed similar anti-apoptotic effects, indicating that R-PC has a direct anti-apoptotic activity. R-PC could improve the apoptosis amorphous of HUVEC cells induced by H2O2. To quantitatively gain insight into the anti-apoptotic effects of R-PC in H2O2-induced HUVEC cells, after treatment with H2O2, the apoptosis rate of HUVEC cells was measured by Annexin-V/PI staining. As shown in Figure 4B, the apoptosis rate grew from 6.09±0.4% to 75.69±2.2% (versus the untreated group). By contrast, resveratrol (10 mg/ml, 20 mg/ml, 40 mg/ml) could evidently attenuate the apoptosis of HUVEC cells to 23.44±1.23%, 35.89±0.31%, 42.34±3.24%, respectively. R-PC (10 mg/ml, 20 mg/ml, 40 mg/ml) could remarkably attenuate the apoptosis of HUVEC cells to 54.05±4.6%, 49.09±2.7% and 46.68±3.1%, respectively. In which 40 mg/mL R-PC reduced the apoptosis remarkably (versus H2O2 alone group).
Page 6, Line 211~:
The entry of quiescent cells into the cell cycle is a critical event in cell growth. To evaluate whether R-PC influenced cell cycle progression in HUVEC cells, cells were treated with R-PC, and then the cell cycle progression of H2O2-treated HUVEC cells in the absence of R-PC using flow cytometry and western blot. The percentage of HUVEC cells in the S and G2/M phases were increased by treatment with H2O2 compared to treatment with control (S phase: 6.33±3.2% in control vs. 12.72±3.4% in H2O2, and G2/M: 23.16 ± 3.8% in control vs. 27.91±2.2% in H2O2). The percentages of cells in the G0/G1 phase were reduced (68.37±3.7% in control vs. 49.97±2.9% in H2O2), indicating that H2O2 induced cell cycle arrest at the G2/M phase.
Page 7, Line 248~:
Weight changes in drosophila can be regarded as a pointer to feed intake. Although, the weight of the female experimental groups did not change remarkably compared with the control group. It showed that the weight of female flies was not decreased through R-PC (Figure. 6A). However, the male experimental groups had a remarkable weight added after feeding R-PC (20 mg/mL, 25 mg/mL, 30 mg/mL) or resveratrol. In addition, female flies showed the most significant improvement in the 30 mg/mL R-PC group, with the average lifespan increasing from 50.2 days to 73.4 days and the maximum lifespan added from 59.1 days to 77.2 days, of which were remarkable (Figure. 6B-D). These data suggested that R-PC showed good longevity effects on female flies.
Page 11, Line 364:
Once caspase-3 is in action, apoptotic cell death is inevitable. The study demonstrated that C-phycocyanin protects chondrocytes from osteoarthritis (OA) progression by reducing ROS production, caspase-3 activity, and cell apoptosis [41].
Page 11, Line 377~:
Further analysis showed that R-PC significantly up-regulated Bcl-2/Bax ratio and down-regulated levels of cytochrome C, which suggests the involvement of the mito-chondrial apoptosis pathway. Moreover, evidence has indicated that apoptosis involves the release of cytochrome C from mitochondria, causing caspase activation. In this study, R-PC decreased the expression of Caspase-3 and PARP, which reduced the H2O2-induced HUVEC cells’ apoptosis (Figure 4, 5). Meantime, R-PC also reversed p21, p27, Cyclin D1 and CDK1/2 expressions after stimulation with H2O2. Our findings suggest that R-PC prevents the H2O2-induced cells cycle phase arrest by regulating cycle-related protein, and apoptosis through mitochondrial and Caspase-3-dependent apoptosis pathways in HUVEC cells.
- Comment 2: Figure 1: 2 of the 3 colors are very similar. Figure 1D caption is missing.
- Response 2: Thanks for reviewer’s kind suggestion. Those comments are all valuable and very helpful for revising and improving our paper, as well as the important guiding significance to our researches. In figure1, black absorption peak means the phycobili protein extract before gel chromatography separation. Red absorption peak showed that the phycobili protein extract was separated by Sephadex G-150 gel chromatography with the R-PC absorbance at 280 nm and at 499 nm, 545 nm and 615 nm. The characteristic absorption peaks of phycoerythrin decreased significantly, while the characteristic absorptions of R-PC increased significantly at 545 nm and 615 nm. However, a weak phycoerythrin characteristic absorption can still be observed at 499 nm, and the ratio of A615/A499 was about 3, indicated that the obtained R-PC sample still contained a small amount of RPE, which needed further purification. Blue absorption peak showed that after purification by DEAE-Sepharose FF chromatography, the obtained R-PC sample only showed the char-acteristic absorption peaks of R-PC at 545 nm and 615 nm, and the ratio of A615/A499 in-creased to 8, indicated that R-PC was further purified. According to reviewer’s comment, The inserted explanations have been added and the caption have been added in the revised manuscript as follow:
Page 3, Line 83 ~:
As shown in Figure 1A, the phycobili protein extract was separated by Sephadex G-150 gel chromatography with the R-PC absorbance at 280 nm, 499 nm, 545 nm and 615 nm. The characteristic absorption peaks of phycoerythrin decreased significantly, while the characteristic absorptions of R-PC increased significantly at 545 nm and 615 nm. The result indicated that a large amount of phycoerythrin and some leucoproteins, which were absorbed at 280 nm had been removed by Sephadex G-150 gel chromatography. However, a weak phycoerythrin characteristic absorption can still be observed at 499 nm, and the ratio of A615/A499 was about 3, indicating that the obtained R-PC sample still contained a small amount of RPE, which needed further purification. After purification by DEAE-Sepharose FF chromatography, the obtained R-PC sample only showed the characteristic absorption peaks of R-PC at 545 nm and 615 nm, and the ratio of A615/A499 increased to 8, indicating that R-PC was further purified.
Page 3, Line 114 ~:
Figure 1. Isolation and purification of R-PC. (A) The absorption spectra of the porphyra haitanensis extract, the RPC fractions from the Sephadex G-150 gel filtration and the DEAE-Sepharose FF chromatography. (B) The gradient SDS-PAGE of the purified RPC. (C) The denaturing-IEF of α subunit of the purified R-PC. (D) The native-IEF of the R-PC purified by the ion exchange chro-matography.
- Comment 3: Figure 2 : axis are not readable.
- Response 3: Thanks for reviewer’s kind suggestion. Those comments are all valuable and very helpful for revising and improving our paper, as well as the important guiding significance to our researches. According to reviewer’s comment, we have revised the Figures of the manuscript to present higher-quality images. The relevant illustrations were added in revised figure 2 as follow:
Revised Figure2:
- Comment 4: Figure 3 : panel C needs a quantification.
- Response 4: Thanks for reviewer’s kind suggestion. Those comments are all valuable and very helpful for revising and improving our paper, as well as the important guiding significance to our researches. According to reviewer’s comment, The quantification were added in revised figure 3C as follow:
Revised Figure3:
- Comment 5: Figure 4 : too small. How the results for 20 mg/mL Resveratrol and R-PC could be explained.
- Response 5: Thanks for reviewer’s kind suggestion. Those comments are all valuable and very helpful for revising and improving our paper, as well as the important guiding significance to our researches. Resveratrol, a natural compound extracted from the skins of grapes, berries, or other fruits, has been shown to have anti-aging, anti-apoptotic and anti-oxidative effects. Resveratrol (10 mg/ml, 20 mg/ml, 40 mg/ml) could evidently attenuate the apoptosis of HUVEC cells to 23.44±23%, 35.89±0.31%, 42.34±3.24%, respectively. R-PC also revealed similar anti-apoptotic effects, indicating that R-PC (10 mg/ml, 20 mg/ml, 40 mg/ml) has a direct anti-apoptotic activity. According to reviewer’s comment, we have been added the inserted explanations and revised the Figures of the manuscript to present higher-quality images as follow:
Revised Figure4:
Page 2, Line 171 ~:
A microscopic fluorescence assay revealed the uniform amorphous of normal HUVEC cell’s nucleus and well-distributed deep blue fluorescence. H2O2-induced apoptosis cells showed typical apoptotic morphology, including broken membranes exhibiting nuclear red fluorescence in cells (Figure. 4A). Resveratrol, a natural compound extracted from the skins of grapes, berries, or other fruits, has been shown to have anti-aging, anti-apoptotic and anti-oxidative effects. R-PC also revealed similar anti-apoptotic effects, indicating that R-PC has a direct anti-apoptotic activity. R-PC could improve the apoptosis amorphous of HUVEC cells induced by H2O2. To quantitatively gain insight into the anti-apoptotic effects of R-PC in H2O2-induced HUVEC cells, after treatment with H2O2, the apoptosis rate of HUVEC cells was measured by Annexin-V/PI staining. As shown in Figure 4B, the apoptosis rate grew from 6.09±0.4% to 75.69±2.2% (versus the untreated group). By contrast, resveratrol (10 mg/ml, 20 mg/ml, 40 mg/ml) could evidently attenuate the apoptosis of HUVEC cells to 23.44±1.23%, 35.89±0.31%, 42.34±3.24%, respectively. R-PC (10 mg/ml, 20 mg/ml, 40 mg/ml) could remarkably attenuate the apoptosis of HUVEC cells to 54.05±4.6%, 49.09±2.7% and 46.68±3.1%, respectively. In which 40 mg/mL R-PC reduced the apoptosis remarkably (versus H2O2 alone group).
- Comment 6: Figure 5 : the quality of the western blot in panel B is of extremely bad quality and cannot be quantified.
- Response 6: Thanks for reviewer’s kind suggestion. Those comments are all valuable and very helpful for revising and improving our paper, as well as the important guiding significance to our researches. According to reviewer’s comment, we have revised the Figures of the manuscript to present higher-quality images. The relevant illustrations were added in revised figure 5 as follow:
Revised Figure5:
- Comment 7: Figure 5 : Figures 6 and 7 : names on the axis are too small.
- Response 7: Thanks for reviewer’s kind suggestion. Those comments are all valuable and very helpful for revising and improving our paper, as well as the important guiding significance to our researches. According to reviewer’s comment, we have revised the Figures of the manuscript to present higher-quality images. The relevant illustrations were added in revised figure 5-7 and moved figure 7 D-E-F to an additional figure as follow:
Revised Figure5:
Revised Figure6:
Revised Figure7:
Revised Figure8:

Round 2
Reviewer 2 Report
The authors improved a lot their article. However I still have some major comments . In figure 2, it is stil leery hard to distinguish the curves. The line has to be be deeper to better see the colors. Microscopic fluorescence pictures (Fig 4A) are still of poor quality. The WB in Fig 5 (the one of Cyclin B1) has to be improved for quantification (there is clearly some bubbles in the middle of the WB that ruins the quantification). The WB has to be re-run.
Author Response
Dear Editor:
Thank you for your letter and for the reviewers’ comments concerning our manuscript entitled “Anti-aging effects of R-phycocyanin from porphyra haitanensis on HUVEC cells and drosophila melanogaster” (ID: marinedrugs-1807399). Those comments are all valuable and very helpful for revising and improving our paper, as well as the important guiding significance to our researches. We have studied comments carefully and have made correction which we hope meet with approval. In addition, we have edited and corrected all typos in manuscript by language editing and native. Because the most current version has added some words, this caused the line numbers has been changed. The main corrections in the paper and the responds to editor’s and review’s comments are in following next page.
Thank you and best regards.
Sincerely,
Yi Zhang, Ph.D. Professor:
Fujian Agriculture and Forestry University, Fuzhou, Fujian, 350000, China.
E-mail: [email protected]
